# Suppressed Hepatic Production of Indoxyl Sulfate Attenuates Cisplatin-Induced Acute Kidney Injury in Sulfotransferase 1a1-Deficient Mice

**DOI:** 10.3390/ijms22041764

**Published:** 2021-02-10

**Authors:** Nozomi Yabuuchi, Huixian Hou, Nao Gunda, Yuki Narita, Hirofumi Jono, Hideyuki Saito

**Affiliations:** 1Department of Clinical Pharmaceutical Sciences, Graduate School of Pharmaceutical Sciences, Kumamoto University, 1-1-1 Honjo, Chuo-ku, Kumamoto 860-8556, Japan; 178y2004@st.kumamoto-u.ac.jp (N.Y.); 202y2051@st.kumamoto-u.ac.jp (H.H.); 158p1012@st.kumamoto-u.ac.jp (N.G.); y-nari@kumamoto-u.ac.jp (Y.N.); hjono@kuh.kumamoto-u.ac.jp (H.J.); 2Department of Pharmacy, Kumamoto University Hospital, 1-1-1 Honjo, Chuo-ku, Kumamoto 860-8556, Japan

**Keywords:** sulfotransferase 1A1 (SULT1A1), indoxyl sulfate (IS), aryl hydrocarbon receptor (AhR), reactive oxygen species (ROS), acute kidney injury (AKI)

## Abstract

Endogenous factors involved in the progression of cisplatin nephropathy remain undetermined. Here, we demonstrate the toxico-pathological roles of indoxyl sulfate (IS), a sulfate-conjugated uremic toxin, and sulfotransferase 1A1 (SULT1A1), an enzyme involved in its synthesis, in cisplatin-induced acute kidney injury using *Sult1a1*-deficient (*Sult1a1*^-/-^ KO) mice. With cisplatin administration, severe kidney dysfunction, tissue damage, and apoptosis were attenuated in *Sult1a1*^-/-^ (KO) mice. Aryl hydrocarbon receptor (AhR) expression was increased by treatment with cisplatin in mouse kidney tissue. Moreover, the downregulation of antioxidant stress enzymes in wild-type (WT) mice was not observed in *Sult1a1*^-/-^ (KO) mice. To investigate the effect of IS on the reactive oxygen species (ROS) levels, HK-2 cells were treated with cisplatin and IS. The ROS levels were significantly increased compared to cisplatin or IS treatment alone. IS-induced increases in ROS were reversed by downregulation of AhR, xanthine oxidase (XO), and NADPH oxidase 4 (NOX4). These findings suggest that SULT1A1 plays toxico-pathological roles in the progression of cisplatin-induced acute kidney injury, while the IS/AhR/ROS axis brings about oxidative stress.

## 1. Introduction

Nephropathy associated with acute kidney injury (AKI) is a serious adverse side effect of cisplatin (cis-diamminedichloroplatinum (II)), occurring in about one-third of the treated patients [1,2]. AKI is a dose-limiting factor in cisplatin treatment and, a decrease in dosage and/or discontinuation of cisplatin treatment is a clinical issue, especially in patients for whom the treatment is significantly effective. There are many reports about the cellular and molecular mechanisms of cisplatin-induced AKI [3,4,5,6,7]. Cisplatin accumulates in renal proximal tubular epithelial cells at about five times the serum concentration and is most highly concentrated in the proximal tubule S3 segment, which is thought to be a toxico-pathogenic target of cisplatin [8]. Organic cation transporter 2 classified as a solute carrier (SLC) 22 family member localized in the basolateral membrane of renal tubular cells is thought to mediate active cellular uptake of cisplatin, thereby causing proximal tubular cell-specific damage [8,9,10,11]. It is known that various factors such as intracellular inflammatory responses, oxidative stress, DNA damage, and apoptosis are involved in the molecular mechanisms of cisplatin-induced AKI [12]. However, the factor that accelerate the progression and deterioration of cisplatin-induced AKI remains undetermined. We previously reported that a typical sulfate-conjugated uremic toxin, indoxyl sulfate (IS), was closely involved in cisplatin nephrotoxicity in an experimental animal model [13,14]. Morisaki et al. [15] reported that IS accumulation in serum and kidney was markedly elevated in cisplatin-treated rats whereas kidney injury was largely prevented by the administration of AST-120, a charcoal adsorbent that decreases intestinal indole derived from the metabolism of tryptophan by gut microbiota, thereby reducing hepatic production of IS. Furthermore, quercetin, a phytochemical polyphenol, significantly suppressed the serum blood urea nitrogen (BUN) level in cisplatin-treated rats and protected them against kidney injury by decreasing renal tissue accumulation of IS [14]. IS is produced in the liver by CYP2E1- and/or CYP2A6-mediated oxidation of indole, followed by sulfotransferase-mediated sulfonation of indoxyl, which then generates IS. An in vitro assay using rat liver fractions revealed that quercetin had a potent inhibitory effect on the sulfonation of indoxyl, thereby suppressing hepatic production of IS [14]. A previous study showed that sulfotransferase (SULT) 1A1 was responsible for the sulfonation activity of indoxyl, whereas other sulfotransferase isoforms, SULT2A1 and SULT1E1, showed no sulfonation activity for indoxyl [16]. However, there is no in vivo evidence for the toxico-pharmacological roles of SULT1A1-mediated IS production in cisplatin-induced nephropathy. The current study using *Sult1a1*-deficient (*Sult1a1*^-/-^ KO) mice provides the first evidence that SULT1A1 contributes as a key modulator in the progression of cisplatin-induced AKI by producing IS that evokes reactive oxygen species (ROS) through an aryl hydrocarbon receptor (AhR) and downregulation of antioxidant enzymes in the kidney.

## 2. Results

### 2.1. Effect of Cisplatin Treatment on IS Concentration in Serum of WT and Sult1a1^-/-^ (KO) Mice

Cisplatin treatment resulted in marked increases in serum IS levels in WT mice, whereas this increase was significantly suppressed in *Sult1a1*^-/-^ (KO) mice (Figure 1). These results indicated that hepatic SULT1A1 was involved in the production and accumulation of IS during cisplatin-induced kidney injury.

### 2.2. Effect of Cisplatin Treatment on Kidney Function and Damage in WT and Sult1a1^-/-^ (KO) Mice

Cisplatin treatment caused a considerable increase in the levels of BUN and serum creatinine in WT mice (Figure 2a,b), but they were significantly attenuated in *Sult1a1*^-/-^ (KO) mice. Figure 2c shows histological alterations in the kidney of WT and *Sult1a1*^-/-^ (KO) mice after cisplatin treatment. In WT mice, cisplatin caused severe histological tubular injuries with the formation of casts derived from sloughed cells, cellular debris, tubular dilation with widening of the lumen, and atrophy of the tubular epithelium. However, tubular injuries caused by cisplatin treatment were markedly attenuated in *Sult1a1*^-/-^ (KO) mice. 

### 2.3. Effect of Cisplatin Treatment on Apoptosis in WT and Sult1a1^-/-^ (KO) Mice Kidney

We examined renal apoptosis by TdT-mediated dUTP Nick End Labeling (TUNEL) staining. The number of TUNEL-positive cells in the kidney was counted. Cisplatin treatment markedly increased the number of apoptotic cells in the kidneys of WT mice, but the apoptosis was significantly suppressed in the kidneys of *Sult1a1*^-/-^ (KO) mice (Figure 3). 

### 2.4. Effect of Cisplatin Treatment on the mRNA Expression of Oxidative Stress and Inflammation-Related Factors of WT and Sult1a1^-/-^ (KO) Mice

Next, we examined the effects of cisplatin treatment on the expression of various factors associated with inflammation and oxidative stress in the kidney. The mRNA expression of interleukin-6 (IL-6) showed a tendency to be elevated in cisplatin-treated WT mice, and this effect was suppressed in *Sult1a1*^-/-^ (KO) mice but the difference was not significant (Figure 4a). IS has been reported to directly activate AhR and to generate oxidative stress through NADPH oxidase-4 (NOX4) in human umbilical vein endothelial cells [17]. AhR expression tended to increase in the kidney of both WT mice and *Sult1a1*^-/-^ (KO) mice treated with cisplatin (Figure 4b). A similar expression profile was observed for xanthine oxidase (XO) (Figure 4c). Cisplatin treatment increased the expression of heme oxygenase (HO)-1 and quinone oxidoreductase (NQO)-1, both of which are downstream factors of the oxidative stress responsive transcription factor Nrf2 (Figure 4d,e). NADPH oxidase 4 (NOX4) was suppressed by cisplatin treatment in WT mice, whereas this effect was suppressed in *Sult1a1*^-/-^ (KO) mice (Figure 4f). The expressions of superoxide dismutase (SOD) 1, SOD2, glutathione peroxidase 1 (GPx1), and catalase (Cat) were decreased by cisplatin treatment in WT mice (Figure 4g–j). The suppressions of SOD1 and GPx1 were partially inhibited in the kidneys of *Sult1a1*^-/-^ (KO) mice. These results suggest that IS induces oxidative stress by suppressing antioxidant enzyme expression. 

### 2.5. Effect of Treatment with or without Cisplatin and IS on HK-2 Cells

In order to further investigate the role of IS in cisplatin-induced AKI, we examined the cytotoxic effect and reactive oxygen species (ROS) production using an immortalized proximal tubule epithelial cell line from normal adult human kidney, HK-2 cells. As shown in Figure 5, HK-2 cells treated with both cisplatin and IS had significantly lower cell viability than the cells treated with IS alone.

### 2.6. Effect of Treatment with Cisplatin and/or IS on ROS Level in HK-2 Cells

The in vivo data from this study revealed that antioxidant enzyme expression was decreased in cisplatin-treated mouse kidneys. Next, we investigated the ROS levels in HK-2 cells. ROS levels were increased in HK-2 cells treated with IS (Figure 6). Combination treatment with cisplatin and IS resulted in significant higher ROS levels compared to IS treatment alone. In addition, ROS levels increased in a time-dependent manner.

### 2.7. AhR Protein Expression in HK-2 Cells Treated with or without Cisplatin and IS

The in vivo data gathered herein suggested that AhR mRNA expression might increase in the kidneys of cisplatin-treated mice. AhR expression was also examined in HK-2 cells. AhR expression decreased in HK-2 cells treated with IS alone (Figure 7). However, when the cells were treated with both cisplatin and IS, this downregulation was less pronounced. 

### 2.8. AhR, XO, and NOX4 Downregulation Decreased ROS Levels Induced by IS

Next, we investigated the ROS levels in HK-2 cells transfected with siAhR, siXO, and siNOX4. The ROS levels were decreased in HK-2 cells treated with IS alone (Figure 8). Moreover, the ROS levels in HK-2 cells treated with both cisplatin and IS were also decreased by siAhR, siXO, and siNOX4. This result suggests that AhR, XO, and NOX4 are partly involved in the increase in ROS induced by IS.

## 3. Discussion

In this study, to the best of our knowledge, for the first time, we identified that SULT1A1 exacerbates cisplatin-induced kidney injury by mediating the production of IS. Cisplatin elevated the IS concentration in WT mice, but it was decreased in KO mice (Figure 1), indicating that SULT1A1 plays an important role in IS production in cisplatin-induced AKI. By inhibiting sulfate conjugation in the liver, the remaining indoxyl could be glucuronidated by UDP-glucuronyltransferase, thereby generating indoxyl glucuronide (IG) as an alternative metabolic pathway. IG is known to be easily removed by dialysis and/or excreted through glomerular filtration in urine owing to its hydrophilic property and low protein-binding ability; it also exhibits hydroxyl radical-scavenging activity, suggesting that it is less toxic than IS [18]. In addition to the significant decrease in IS accumulation in *Sult1a1*^-/-^ (KO) mice, kidney function and histological damage were significantly reduced (Figure 1 and Figure 2). These results suggest that SULT1A1 plays an important role in the pathology of cisplatin-induced AKI by controlling the production of IS.

It was also observed that renal cell apoptosis was significantly decreased in *Sult1a1*^-/-^ (KO) mice (Figure 3). In vitro data showed that the cell survival rate may be lower in HK-2 cells treated with both cisplatin and IS than in those treated with cisplatin alone (Figure 5). Renal tubular cell death is thought to be a secondary consequence of cellular dysfunction including oxidative stress. Apoptosis and necrosis underlie cell death, which triggers inflammatory processes, accelerating renal tubular injury [19]. The present data demonstrated that IS was an exacerbating factor in cisplatin-induced kidney injury, accompanied by the stimulation tubular apoptosis. 

Oxidative stress is another important factor involved in the exacerbation of cisplatin-induced AKI. Oxidative stress induces excessive ROS accumulation, which is caused by an imbalance between oxidants and antioxidants [20]. Recep et al. reported that the oxidant parameters were increased and that the antioxidant parameters were decreased in cisplatin-induced AKI [21]. We examined various oxidative stress-related factors because the factors intervening between IS accumulation and oxidative stress production in the kidney were not clear. Among the oxidative stress-related factors (Figure 4a–j), it was found that XO, an oxidant enzyme, was upregulated in cisplatin-treated WT and *Sult1a1*^-/-^ (KO) mice (Figure 4c). Meanwhile, the cisplatin-induced upregulation of antioxidant enzymes, SOD1, SOD2, GPx1, and Cat in WT mice may be suppressed in cisplatin-treated *Sult1a1*^-/-^ (KO) mice (Figure 4g,h). This may result in increased ROS accumulation with induced oxidative stress.

On the other hand, AhR expression was elevated by cisplatin treatment both in vivo and in vitro (Figure 4b and Figure 7). A previous report suggested that TGF-β1 increased AhR expression in naïve CD4+ cells [22]. It has also been reported that TGF-β1 expression is increased in cisplatin-induced AKI mouse kidneys [23]. Thus, cisplatin treatment could enhance AhR expression via TGF-β1. In addition, IS treatment decreased AhR expression, suggesting the possibility that IS may bind to AhR as a ligand. A previous study also reported that AhR was ubiquitinated and degraded by the proteasome system after binding to its ligand [24]. It has recently been reported that IS is an endogenous agonist of AhR [25,26,27,28]. AhR is a ligand-activated transcription factor involved in the biological detoxification of ligands [29]. Under normal conditions, AhR is located in the cytoplasm in an inactive state as part of a complex with stabilizing proteins, such as heat-shock protein 90, a co-chaperone p23 (P23), and an X-associated protein 2 [30]. After ligand binding, AhR is activated by a conformational change, followed by phosphorylation by protein kinase C; thus, it translocates to the nucleus as the ligand-bound AhR complex [31]. In the nucleus, this complex releases AhR, which binds to the AhR nuclear translocator (ARNT), forming a ligand-bound AhR-ARNT dimer. This heterodimer binds to specific DNA sequences (referred to as DRE or XRE for dioxin- or xenobiotic-responsive elements) located within the promoters of target genes, such as cytochrome P450 (CYP)1A1, CYP1A2, CYP1B1, cyclooxygenase-2 (COX-2), NQO-1, UDP-glucuronosyltransferase (UGT1a1), and XO/xanthine dehydrogenase [32,33]. Therefore, the present findings suggest that IS may accumulate in the renal tissue and activates AhR, thereby enhancing the expression of oxidative stress-related factors, which leads to an increase in ROS. 

Next, we examined the ROS levels in HK-2 cells. In our study, the ROS levels were significantly increased in HK-2 cells after cisplatin and IS combined treatment, as expected (Figure 6). At the same time, although the expression of AhR was downregulated after treatment with both cisplatin and IS, the most important point was that its expression appeared at a significantly higher level than when treated with IS alone (Figure 7). Because cisplatin upregulated AhR expression, the ROS levels were increased following treatment with both cisplatin and IS.

Recently, Nakagawa et al. reported that IS induced the production of ROS via the AhR-NOX pathway in vascular tissue [34]. AhR is known to transcriptionally activate XO [35]. It has been reported that IS induces ROS by activating NOX4 [36]. Thus, we examined ROS levels in HK-2 cells transfected with siAhR, siXO, or siNOX4 (Figure 8). The ROS levels were decreased in HK-2 cells treated with IS. However, none of the knockdowns completely suppressed ROS in HK-2 cells treated with both cisplatin and IS. ROS are produced mainly in two pathways: one is dependent on the activation of NOX located in the cell membrane, and the other produces ROS as a byproduct of ATP production in the mitochondria. IS is known to induce mitochondrial dysfunction [37]; moreover, cisplatin-induced AKI is linked with mitochondrial dysfunction [38]. Furthermore, ROS production is elevated in damaged mitochondria. IS may also be associated with another NOX isoform and mitochondrial ROS production caused by mitochondrial dysfunction in cisplatin-induced AKI. Taken together, these findings suggest the involvement of SULT1A1 and IS in the progression or deterioration of cisplatin-induced kidney injury via the cisplatin/IS/AhR/ROS axis (summarized in Figure 9).

In summary, although our study is limited to the toxicological roles of IS and its hepatic production enzyme SULT1A1 in cisplatin-induced AKI and has not explored the involvement of other harmful uremic toxins such as p-cresyl sulfate or trimethylamine-N-oxide, our data identified for the first time the toxicological and pathological roles of IS and SULT1A1 in cisplatin-induced AKI. In particular, we identified novel mechanism indicating the involvement of the cisplatin/IS/AhR/ROS axis in producing oxidative stress. IS is involved not only in cisplatin-induced AKI but also as a factor in the development of ischemic AKI and chronic kidney disease (CKD) with renal interstitial fibrosis [39,40,41]. Furthermore, as it has been demonstrated that IS is also involved in heart and vascular tissues as a toxic endogenous factor through the induction of oxidative stress [42], it is important to suppress IS production or accumulation. Thus, hepatic SULT1A1 may be useful as a therapeutic target for diseases in which IS accumulation induces toxic damage or injury.

## 4. Materials and Methods

### 4.1. Reagents

IS was obtained from Sigma-Aldrich Co. (St. Louis, MO, USA). Cisplatin was purchased from Nippon Kayaku (Tokyo, Japan).

### 4.2. Sult1a1-Deficient Mice

*Sult1a1*-deficient mouse embryos (Deltagen, San Carlos, California, USA) were purchased, melted, and transplanted in an expedient parent at the Kumamoto University Institute of Resource Development and Analysis (IRDA), producing the hetero mouse. These mice were bred in the Animal Resources and Development Center (CARD) of Kumamoto University. Homogenized mice were subsequently produced.

### 4.3. Animal Experiments

All procedures for animal experiments were approved by the Kumamoto University animal ethics committee (Identification code: A 2019-046, approval date: 2019), and the animals were treated in accordance with the guidelines of the United States National Institutes of Health regarding the care and use of animals for experimental procedures and with the Guidelines of Kumamoto University for the care and use of laboratory animals. Normal male C57BL/6J (wild-type (WT)) mice were purchased from CLEA Japan, Inc. (Tokyo, Japan). Six-week-old mice were housed in a standard animal maintenance facility at a constant temperature (22 ± 2 °C) and humidity (50%–70%) and a 12/12-h light/dark cycle for approximately one week before the day of the experiment, with food and water available ad libitum. Mice were administered cisplatin or saline (25 mg/kg) by intraperitoneal injection. The serum and tissue samples were collected 72 h after cisplatin administration. Serum BUN, and the creatinine concentrations were measured using the Fuji dry chem Nx500 (Fujifilm, Kanagawa, Japan).

### 4.4. Liquid Chromatography/Mass Spectrometry/MS (LC/MS/MS) Analysis

IS concentrations were measured using an API 3200TM LC/MS/MS system (AB SCIEX, Foster City, CA, USA) with a triple quadruple mass spectrometer following negative ion mode: IS, m/z 212.08; IG, m/z 308.21. The source parameters were optimized to obtain the highest analysis peak. Chromatographic separation was performed on a Symmetry™ C18 (5 μm, 3.9 × 150 mm) column (Waters Corp, Milford, MA, USA). The samples were eluted at a flow rate of 0.2 mL/min using a mobile phase consisting of 10 mM ammonium acetate: acetonitrile (73:27 *v/v*) at 40 °C. The MS/MS operating conditions were as follows: charged aerosol detection gas at 3.0 psi, curtain at 40 psi, ion source gas (GS) 1 at 50 psi, GS2 at 30 psi, IS at −4500 V, and temperature at 500 °C.

### 4.5. HE Staining

Kidney samples were fixed in 10 % neutral-buffered formalin for 24 h and embedded in paraffin. Paraffin-embedded specimens were cut into 2 μm sections and mounted onto glass slides. The sections were deparaffinized and stained with hematoxylin and eosin (HE).

### 4.6. TUNEL Assay

To examine the appearance of apoptotic cells in the kidney, specimens from WT and *Sult1a1^-/-^* (KO) mice were TUNEL-stained using an in situ apoptosis detection kit (Takara Bio Inc., Shiga, Japan). Under light microscopy (200× magnification), the number of TUNEL-positive cells in the area covering the majority of corticomedullary junctions in a slide were counted by researchers blinded to the samples. The number of TUNEL-positive cells in five areas of a section for each mouse was counted. Data from mice in the same treatment group were averaged.

### 4.7. Quantitative PCR Assay

The kidney tissue was homogenized, and total RNA was isolated by a phenol-chloroform extraction method using Trizol (Thermo Fisher Scientific, Waltham, MA, USA). cDNA was generated using the PrimeScript RT reagent kit (Takara Bio Inc., Osaka, Japan) according to the manufacturer’s instructions. Each PCR assay was performed using 1 μL of cDNA and each primer at 0.3 μM in a LightCycler^®^ 480 System (F. Hoffmann-La Roche Ltd., Basel, Switzerland) with TB Green Premix DimerEraserTM (Takara Bio Inc., Shiga, Japan). The primers were purchased from Sigma (St Louis, MO, USA). The primer sequences for each gene are shown in Table 1 below. Amplification was performed using the LightCycler^®^ 480 System (F. Hoffmann-La Roche Ltd., Basel, Switzerland), and each reaction was performed under the following conditions: initialization for 30 s at 95 °C, followed by 50 cycles of amplification, with 5 s at 95 °C for denaturation and 30 s at 72 °C for annealing and elongation. The expression levels were normalized to those of Glyceraldehyde 3-phosphate dehydrogenase (GAPDH).

### 4.8. Cell Cultures

HK-2 cells derived from human proximal tubular cells were cultured in DMEM/F12 with HEPES (GIBCO, Tokyo, Japan) supplemented with 10% fetal bovine serum (FBS) in a 37 °C, humidified incubator with 5% CO_2_.

### 4.9. Analysis of Cell Survival Rate

Cells were incubated in 24-well plates (1.0 × 10^5^ cells/mL) for 24 h and were treated with cisplatin (5 μg/mL) and/or IS (500 μM) in serum-free DMEM/F12 with HEPES. Following exposure, the medium was removed from the wells and the cells were cultured DMEM/F12 with HEPES containing 25 μL CCK-8 (DOJINDO, Kumamoto, Japan) in each well at 37 °C for 1 h. Cell viability was measured using a plate reader.

### 4.10. ROS Analysis

HK-2 cells were incubated in 96-well plates (1.0 × 10^4^ cells/mL) for 24 h, were washed with PBS, and were incubated with 5 μM CM-H_2_DCFDA (Thermo Fisher Scientific, Waltham, MA, USA) for 30 min. After incubation, CM-H_2_DCFDA was removed from the wells and the cells were treated with cisplatin (5 μg/mL) and/or IS (500 μM). Following exposure, fluorescence intensity was measured using a micro plate reader SH-9000 Lab (Hitachi, Tokyo, Japan).

### 4.11. Transfection with Small Interfering RNA (siRNA)

HK-2 cells were incubated in 96-well plates (1.0 × 10^4^ cells/mL) for 24 h and were transiently transfected with AhR (sc-29654, Santa Cruz Biotechnology, Dallas, Texas, USA), XO (sc-41691, Santa Cruz Biotechnology, Dallas, Texas, USA), and NOX4 (sc-41586, Santa Cruz Biotechnology, Dallas, Texas, USA) siRNA (50 nM) using Lipofectamine RNAi MAX (Thermo Fisher Scientific, Waltham, MA, USA) according to the manufacturer’s protocol. After transfection and incubation for 48 h, the experiments were performed. Silencer negative control siRNA (Thermo Fisher Scientific, Waltham, MA, USA) was used as the control (siControl).

### 4.12. Western Blotting

Cells were lysed in ice-cold lysis buffer. After measuring the protein content using the bicinchoninic acid assay (Thermo Fisher Scientific, Waltham, MA, USA), each sample was mixed with loading buffer (2% *w/v* sodium dodecyl sulfate (SDS), 125 mM Tris-HCl pH 7.2, 20% *v/v* glycerol, and 5% *v/v* 2-mercaptoethanol) and incubated at 95 °C for 2 min. The samples were subjected to sodium dodecylsulfate-polyacrylamide gel electrophoresis using a 7.5% gel and transferred onto a polyvinylidene difluoride membrane (Immobilon-P; EMD Millipore, Billerica, MA, USA) by semi-dry electroblotting. The membrane was blocked for 1 h at room temperature with 5%*v/v* non-fat dry milk (Cell Signaling Technology, Denver, CO, USA) in 50 mM Tris-buffered saline (pH 7.6) containing 0.3%*v/v* Tween 20 and then incubated overnight at 4 °C with a primary antibody specific for AhR (sc-133088, Santa Cruz Biotechnology, Dallas, Texas, USA) or β-actin (A5441, Sigma, St. Louis, MO, USA). The blots were then washed with Tris-buffered saline containing Tween 20 and incubated with secondary antibody (NA-931, GE Healthcare Ltd., Chicago, IL, USA) for 1 h at room temperature. Immunoblots were visualized with an ECL system (ECL Advance Western Blotting Detection Kit; GE Healthcare Ltd., Chicago, IL, USA).

### 4.13. Statistical Analysis

The data were analyzed statistically by analysis of variance, followed by Tukey–Kramer or Scheffe’s multiple comparison test. A *p*-value of <0.05 was considered statistically significant. All data are presented as the mean ± standard deviation (SD).

## Figures and Tables

**Figure 1 ijms-22-01764-f001:**
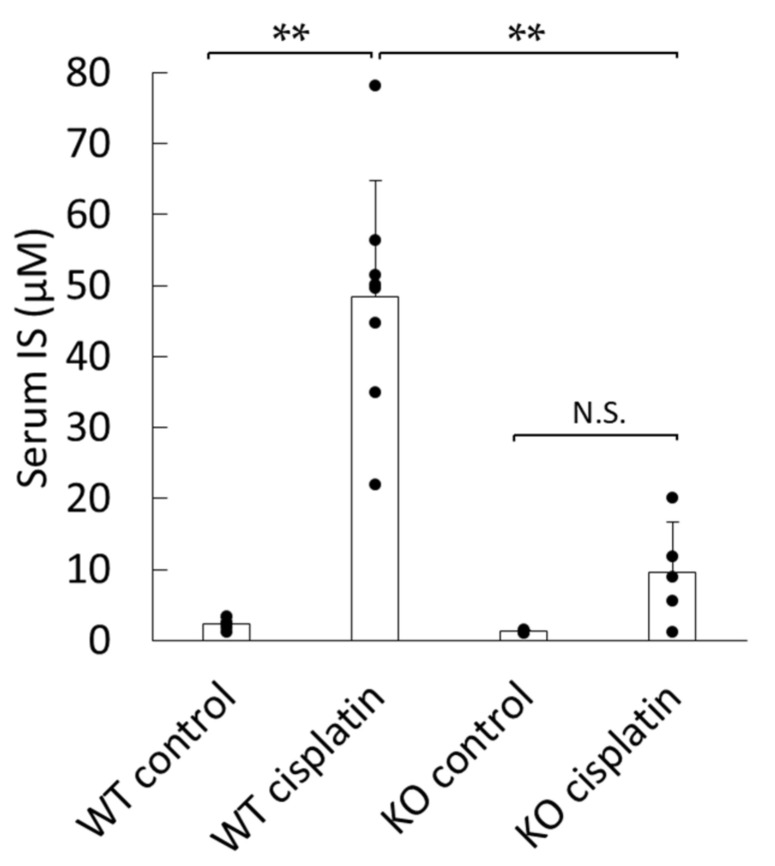
Indoxyl sulfate (IS) concentration in the serum of wild-type (WT) and *Sult1a1*^-/-^ (KO) mice 72 h after treatment with saline or cisplatin: each column represents the mean ± S.D. of 4–8 mice. ** *p* < 0.01; N.S.: not significant.

**Figure 2 ijms-22-01764-f002:**
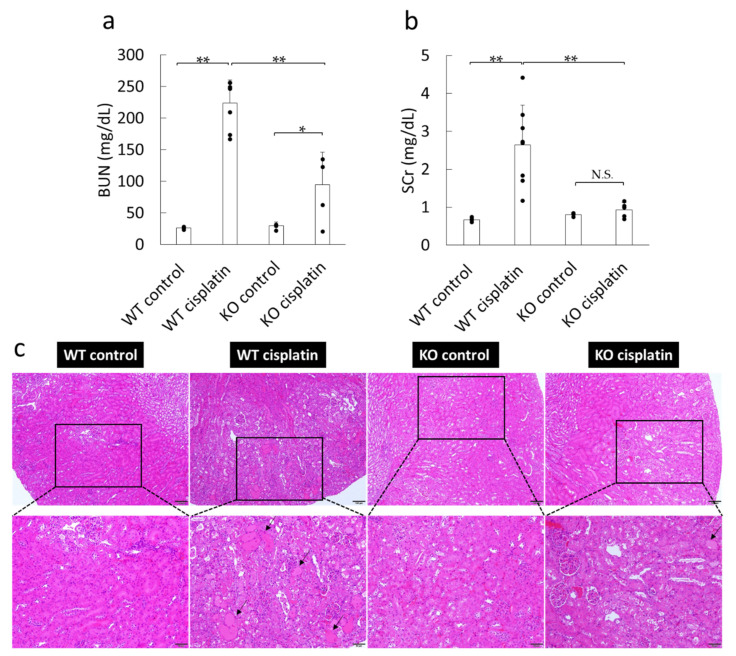
Effect of cisplatin treatment on kidney function and damage in WT and *Sult1a1*^-/-^ (KO) mice: (**a**,**b**) the effect of cisplatin treatment on (**a**) BUN and (**b**) serum creatinine, where each column represents the mean ± S.D. of 4–8 mice, * *p* < 0.05, ** *p* < 0.01, and N.S. represents not significant, and (**c**) the effect of cisplatin treatment on kidney tissue histology (hematoxylin and eosin (HE) staining). The arrows indicate tubular cast deposition. Scale bar, 100 μm (**up**), 50 μm (**down**).

**Figure 3 ijms-22-01764-f003:**
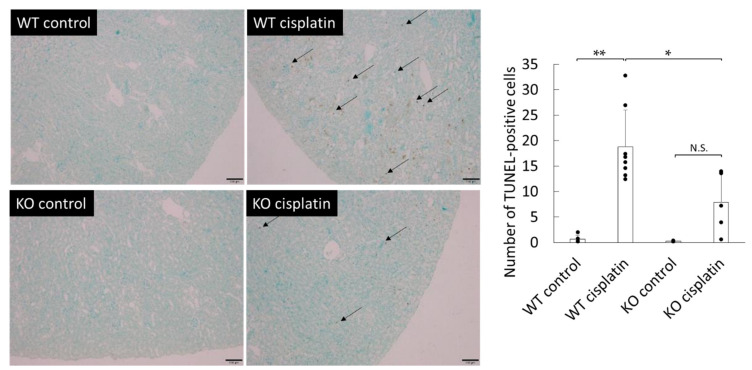
Detection of TUNEL-positive cells in the kidney of WT and *Sult1a1*^-/-^ (KO) mice: the arrows indicate typical apoptotic cells. Scale bar, 100 μm. Each column represents the mean ± S.D. of 4–8 mice. * *p* < 0.05; ** *p* < 0.01; N.S., not significant.

**Figure 4 ijms-22-01764-f004:**
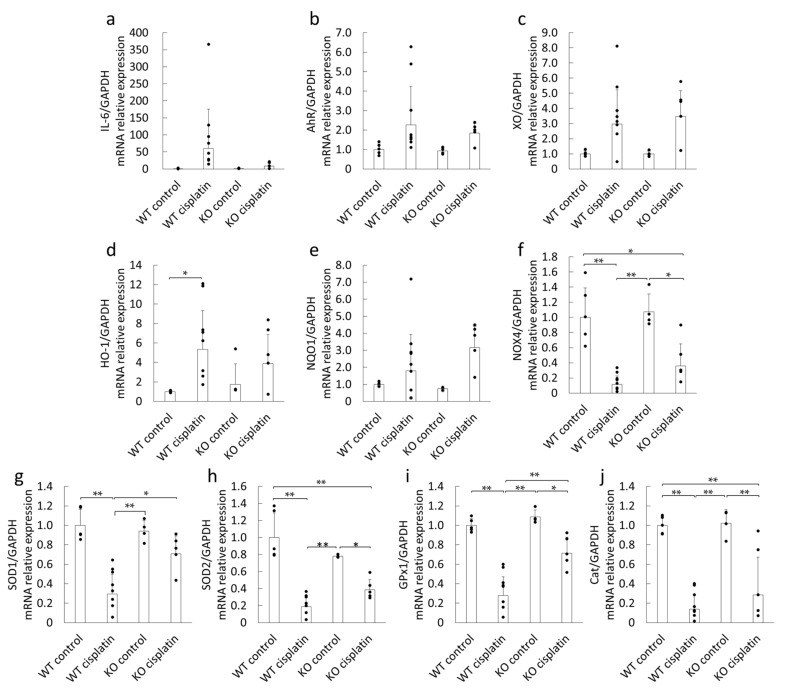
Effect of cisplatin treatment on mRNA expression of (**a**) interleukin-6 (IL-6), (**b**) aryl hydrocarbon receptor (AhR), (**c**) xanthine oxidase (XO), (**d**) heme oxygenase-1 (HO-1), (**e**) quinone oxidoreductase-1 (NQO1), (**f**) NADPH oxidase 4 (NOX4), (**g**) superoxide dismutase (SOD) 1, (**h**) SOD2, (**i**) glutathione peroxidase 1 (GPx1), and (**j**) catalase (Cat) in the kidneys of WT and *Sult1a1*^-/-^ (KO) mice: each column represents the mean ± S.D. of 4–8 mice. * *p* < 0.05; ** *p* < 0.01; N.S., not significant.

**Figure 5 ijms-22-01764-f005:**
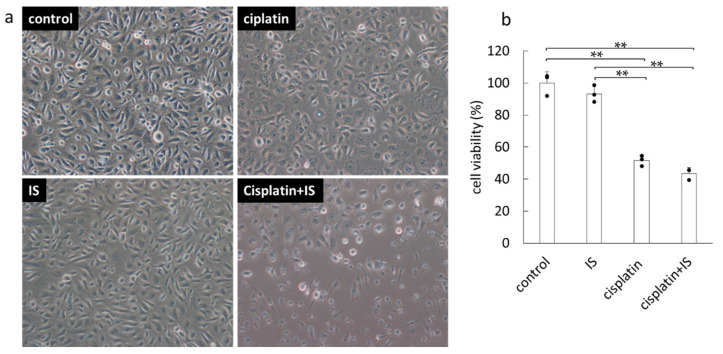
Effect of treatment with cisplatin and/or IS on HK-2 cells: (**a**) appearance of HK-2 cells and (**b**) cell survival rates after 24 h treatment with or without cisplatin (5 µg/mL) and/or IS (500 μM). Each column represents the mean ± S.D. of triplicate samples. ** *p* < 0.01; N.S., not significant.

**Figure 6 ijms-22-01764-f006:**
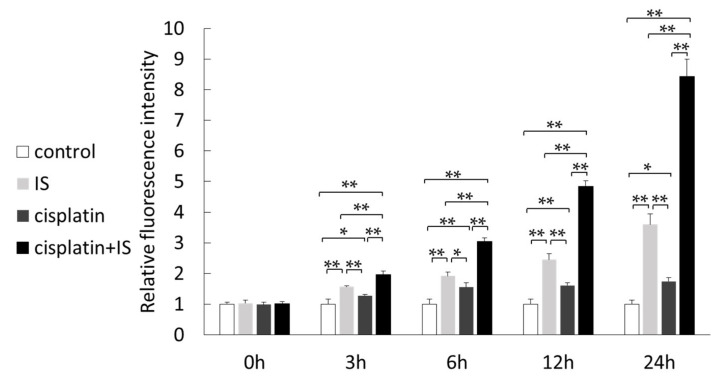
Effect of treatment with cisplatin and/or IS on the reactive oxygen species (ROS) level in HK-2 cells: the data were corrected with the control for each time point. Each column represents the mean ± S.D. of triplicate samples. * *p* < 0.05; ** *p* < 0.01.

**Figure 7 ijms-22-01764-f007:**
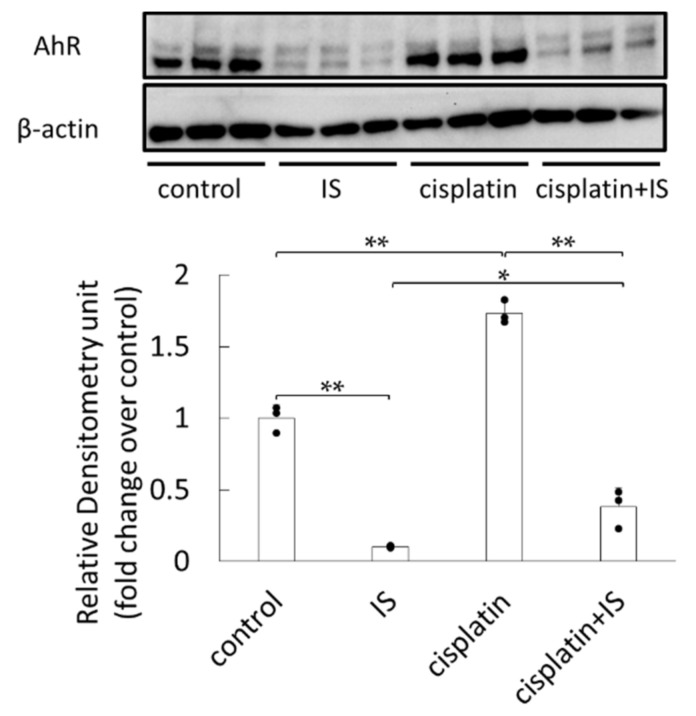
Effect of treatment with cisplatin and/or IS on AhR expression in HK-2 cells: each column represents the mean ± SD of triplicate samples. * *p* < 0.05; ** *p* < 0.01.

**Figure 8 ijms-22-01764-f008:**
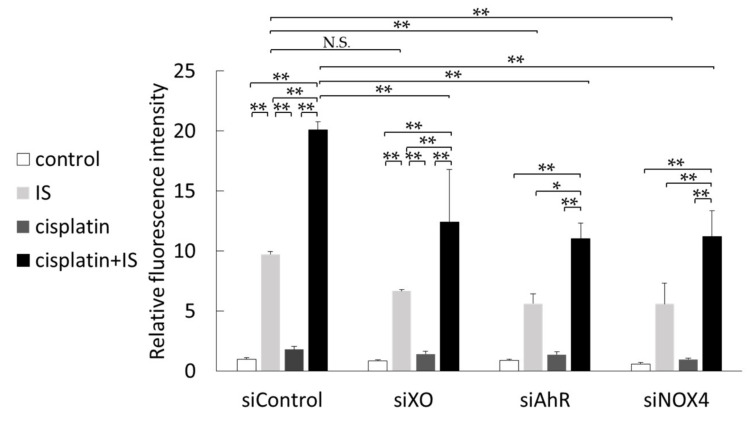
Effect of treatment with cisplatin and/or IS on ROS levels in HK-2 cells with or without knockdown of AhR, XO, and NOX4: the data were corrected with the control siControl. Each column represents the mean ± SD of triplicate samples. * *p* < 0.05; ** *p* < 0.01; N.S., not significant.

**Figure 9 ijms-22-01764-f009:**
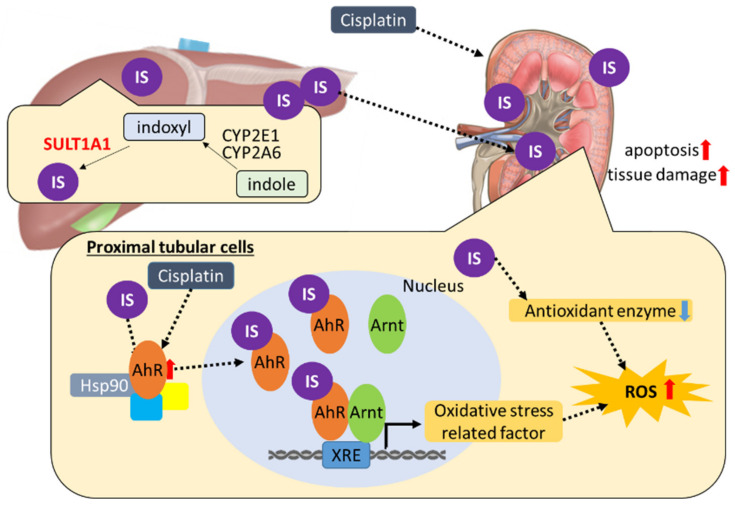
Proposed schematic representation summarizing the toxicological involvement of IS in the pathogenesis mechanism of acute kidney injury induced by cisplatin: with accumulation of serum IS, exposure the kidney to IS increases. In renal tubular cells, cisplatin partially enhances AhR expression. IS interacts as a ligand with a AhR complex composed of heat-shock protein 90 and a 43 kDa protein, triggering a conformational change in AhR to a form exhibiting higher affinity for DNA. The IS–AhR complex translocate to the nucleus and binds specific DNA sequences called XRE (xenobiotics response element) by forming a heterodimer with Arnt, thereby activating gene expression that leads to oxidative stress. IS accumulates in renal tissue and decreases antioxidant enzymes that may be involved in ROS elevation. ROS elevation may result in apoptosis and tissue damage.

**Table 1 ijms-22-01764-t001:** The primer sequences for each gene

Gene	Forward (5′–3′)	Reverse (5′–3′)
IL-6	TACCACTTCACAAGTCGGAGGC	CTGCAAGTGCATCATCGTTGTTC
GPx1	CGCTCTTTACCTTCCTGCGGAA	AGTTCCAGGCAATGTCGTTGCG
Cat	CGGCACATGAATGGCTATGGATC	AAGCCTTCCTGCCTCTCCAACA
HO-1	AACAAGCAGAACCCAGTCTATGC	AGGTAGCGGGTATATGCGTGGGCC
NQO-1	AGGGTTCGGTATTACGATCC	AGTACAATCAGGGCTCTTCTCG
NOX4	CGGGATTTGCTACTGCCTCCAT	GTGACTCCTCAAATGGGCTTCC
SOD1	GGTGAACCAGTTGTGTTGTCAGG	ATGAGGTCCTGCACTGGTACAG
SOD2	TAACGCGCAGATCATGCAGCTG	AGGCTGAAGAGCGACCTGAGTT
XO	GCTCTTCGTGAGCACACAGAAC	CCACCCATTCTTTTCACTCGGAC
AhR	AGGCTAAGAGAGCCTTGTCT	TCCAACACTTTCTGGACAGG
GAPDH	CGACTTCAACAGCAACTCCCACTCTTCC	TGGGTGGTCCAGGGTTTCTTACTCCTT

## Data Availability

The data that support the findings of this study are available from the corresponding author upon reasonable request.

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
