# Peer review of "Suppressed Hepatic Production of Indoxyl Sulfate Attenuates Cisplatin-Induced Acute Kidney Injury in Sulfotransferase 1a1-Deficient Mice"

_ijms, 2021, doi:10.3390/ijms22041764_

Round 1

Reviewer 1 Report

The manuscript entitled:” Suppressed hepatic production of indoxyl sulfate attenuates 2 cisplatin-induced acute kidney injury in sulfotransferase 1a1-3 deficient mice.”, by Yabuuchi et al. investigates the toxico-pathological roles of indoxyl sulfate (IS) and sulfotransferase 1A1 (SULT1A1), an enzyme involved in its synthesis, in cisplatin-induced acute kidney injury using Sult1a1-deficient mice.

Specific remarks

-It puzzles the reviewer how preventing of sulfation could lead to a decrease in toxic effects. Sulfation of metabolites and drug is seen as a detoxification process and improves solubility of the metabolites facilitating their removal. If compounds like indoxyl and others such as p-cresol are not sulfated at the side of the liver,  do these compounds then accumulate in the circulation of these AKI KO mice? I would expect this to give even more toxicity…Please discuss

It would be useful to report indole or indoxyl levels next to IS

-In the description of the results tha authors write that :” Although cisplatin treatment in Sult1a1-/- (KO) mice also resulted in elevated levels of BUN and serum creatinine, these increases were significantly attenuated compared to those in WT mice.” According to figure 2B serum creatinine is not increase in the KO mice. Please explain.

-the results under 2.4 are not well described in view of their statistical significance, cfr. …may be elevated…might be suppressed… It would be better to describe this a ther is a trend , although not significant…

Effect of cisplatin on NQo1 is not significant according to the figure 4e

“Downregulation of SOD1 and GPx1 expression were suppressed in the kidneys of Sult1a1-/- (KO) mice.” Better suppression of SOD1 and GPx1 was partially inhibited in the kidneys of Sult1a1-/- (KO)

-under 2.5 cisplatin decreased cell viability, however there was not additive effect of IS, please adjust description

-under 2.6 since at 24h in the presence of cisplatin, only half of the HK-2 cells were viable, it should be stressed that the RFI was measured in the viable cells only. Were results normalised for the number of viable cells? (this is not described in the methods.

-under 2.7: “However, when the cells were treated with both cisplatin and IS, this downregulation was ameliorated.” Better to say that it was less pronounced.

-Please replace “renal” by “kidney” throughout the text

Reviewer 2 Report

Your manuscript is written about the dysfunction of sulfotransferase 1a1 attenuates cisplatin-induced acute kidney injury. Your study has an interesting and important information for medication in patients with taking cisplatin, but I have some important comments for your manuscript.

- Comments -

  1. I understand your results in that hepatic production of indoxyl sulfate by SULT1A1 plays important roles in cisplatin-induced acute kidney injury. I had thought that indoxyl sulfate accumulates by kidney dysfunction in patients with taking CDDP. However, your results (Figure 2B etc.) suggest that accumulation of indoxyl sulfate in patients with taking CDDP is induced by other mechanism other than kidney dysfunction. I think that your discussion about it (Reversal of causality) will be a better manuscript. In Figure 1, the result that cisplatin tends to increase serum concentration of indoxyl sulfate in KO mice is so interesting in relation to the above comments.

  1. I think that indoxyl sulfate concentration in in vitro experiments is too high in comparison with unbound plasma concentration in patients with kidney dysfunction.

  1. Your manuscript is not well-formatted as a manuscript. First, I think that you do not have to discuss and include reference in Results section. Second, Discussion is not an overview. I think that you have to transfer some sentence in Discussion section to Introduction section and Materials and Methods section.

Round 2

Reviewer 1 Report

In the revised version of the manuscript entitled: ”Suppressed hepatic production of indoxyl sulfate attenuates cisplatin-induced acute kidney injury in sulfotransferase 1a1-3 deficient mice.”, by Yabuuchi et al. the authors addressed my comments.

To address my first comment on “how preventing of sulfation could lead to a decrease in toxic effects since sulfation of metabolites and drug is seen as a detoxification process which improves solubility of the metabolites, facilitating their removal”,the authors added an interesting thought to the discussion that “By inhibiting sulfate conjugation in the liver, remained indoxyl could be glucuronidated by UDP-glucuronyltransferase, thereby generating indoxyl glucuronide (IG) as an alternative metabolic pathway (unpublished observations in our in vitro study using hepatic S9 fraction). IG is known to be removed easily by dialysis and/or excreted through glomerular filtration into urine as its hydrophilic property and low protein binding.”

With this thought the authors induce the interest in quantification of IG levels in their Sult1a1-/- KO model, to support this statement, but IG levels are currently not presented.

The reviewer still would like to know if the authors have IG levels available.

Reviewer 2 Report

I think that you have revised your manuscript appropriately.
